# Shift in Potential Malaria Transmission Areas in India, Using the Fuzzy-Based Climate Suitability Malaria Transmission (FCSMT) Model under Changing Climatic Conditions

**DOI:** 10.3390/ijerph16183474

**Published:** 2019-09-18

**Authors:** Soma Sarkar, Vinay Gangare, Poonam Singh, Ramesh C. Dhiman

**Affiliations:** ICMR-National Institute of Malaria Research, Dwarka sector 8, Delhi 110077, India; ssarkar.delhi@gmail.com (S.S.); vgangare@gmail.com (V.G.); punamsingh10@gmail.com (P.S.)

**Keywords:** climate suitability, climate change, malaria, fuzzy sets, cordex data

## Abstract

The future implications of climate change on malaria transmission at the global level have already been reported, however such evidences are scarce and limited in India. Here our study aims to assess, identify and map the potential effects of climate change on *Plasmodium vivax* (*Pv*) and *Plasmodium falciparum* (*Pf*) malaria transmission in India. A Fuzzy-based Climate Suitability Malaria Transmission (FCSMT) model under the GIS environment was generated using Temperature and Relative Humidity data, extracted from CORDEX South Asia for Baseline (1976–2005) and RCP 4.5 scenario for future projection by the 2030s (2021–2040). National malaria data were used at the model analysis stage. Model outcomes suggest that climate change may significantly increase the spatial spread of Pv and Pf malaria with a numerical increase in the transmission window’s (TW) months, and a shift in the months of transmission. Some areas of the western Himalayan states are likely to have new foci of Pv malaria transmission. Interior parts of some southern and eastern states are likely to become more suitable for Pf malaria transmission. Study has also identified the regions with a reduction in transmission months by the 2030s, leading to unstable malaria, and having the potential for malaria outbreaks.

## 1. Introduction

Malaria is still a major public health challenge in India, where around one million cases are reported annually [1]. One of the major reasons for the persistence of malaria is the extensive geographic and climatic diversity of the country, which supports ideal ecological conditions for sustaining the parasites and their vectors. Studies have found that the spatial limits of the distribution and seasonality of malaria are sensitive to the seasonal characteristics of climatic factors [2,3,4]. The major climatic determinants of malaria are temperature, rainfall and humidity [5,6,7,8,9]. Impact of climate change is not uniform around the globe: Some places may become warmer and drier, while others warmer and wetter [10]. Hence, the threat of climate change is expected to have a profound effect on the mosquito’s longevity, development of malaria parasites in the vectors, and consequently opening the windows of malaria transmission particularly in areas which are free due to temperature constrains. In other words, global climate change is likely to alter the spatial and temporal distribution of malaria.

Prior studies suggest that climate change will increase the opportunities for malaria transmission in traditionally non-malarious areas, and make it difficult to control in traditionally malarious areas due to an alteration in their growth cycle and transmission seasons [11,12,13]. Studies in recent decades have reported the evidences about an increase in the spread of the disease in the current malaria endemic areas [14,15], a reemergence of the disease in areas which have eliminated the disease in the past [16], and at the same time, areas with reduced endemicity due to reduction in overall vectorial capacity [17]. Although it has been recognized that the direct and indirect effects of climate change depend upon the population’s ability (social, economic and political environment) to cope with and respond to disease burdens, the impacts of future climate change on malaria transmission cannot be ignored [18,19].

The future repercussions of climate change on malaria transmission at the global level have already been explored; however, such evidences are limited in India. In India, climate suitability models for malaria transmission were developed mainly by considering the Temperature and Rainfall variables [20,21]. Based on PRECIS data (baseline of 1961–1990) for Temperature and Relative Humidity (RH), Dhiman et al., 2011 [22] had projected the climate suitability of malaria transmission with respect to climate change by the year 2030. These regional models were developed on traditional threshold-based hard partitioning. Soft partitioning approaches like Fuzzy logic-based climate suitability models have been applied to define suitable and unsuitable areas for malaria transmission in many studies [3,23,24,25,26] around the globe, but not in India. While resolving the uncertainty in defining distinct thresholds of most suitable to least suitability, the present study adopts the soft partitioning approach using Temperature and RH to map potential malaria transmission vulnerability in the context of climate change. Hence, a Fuzzy-based Climate Suitability Malaria Transmission (FCSMT) model using CORDEX (Coordinated Regional Climate Downscaling Experiment) data is generated with the objective of identifying climate-based *Plasmodium vivax* (*Pv*) and *Plasmodium falciparum* (*Pf*) malaria transmission vulnerability with respect to the climate change scenario. Resultant maps would guide the National Vector Borne Disease Control Programme (NVBDCP) in addressing the early preparedness to eradicate malaria. The near-future projected malaria transmission scenario in the present study is limited to the 2030s for two reasons: One, climate sensitivity uncertainty increases in the long term projections; and two, this study is an initiative towards the Government of India’s National Framework for Malaria Elimination in India 2016–2030 Program [27].

## 2. Materials and Methods

### 2.1. Characteristics of Malaria Transmission in India

In India, the distribution of vectors and malaria endemicity largely depends on the physiographic, climatic, eco-epidemiology and socio-developmental conditions and varying from area to area. The northern part of the country is characterized by a subtropical climate and increase in altitude towards north, and has high variation of temperature between summer and winter with only a brief rainy season, providing suitable RH for a shorter period. The southern part of India has a tropical climate and is surrounded by seas; the temperature varies little throughout the year, and receives significant SW and NE monsoonal rains. The northeastern states are characterized by the Himalayan ranges receiving heavy annual rainfall (>2000 mm), a most conducive temperature, and thus stable malaria transmission. The western states, particularly Rajasthan, Gujarat and parts of Karnataka, are mainly plain areas, receive scanty rainfall (<1000 mm), and are prone to malaria outbreaks.

There are nine major vectors of malaria in India, of which *An. culicifacies*, *An. fluviatilis*, *An. stephensi*, *An. minimus* and *An. dirus* are the primary vectors in India, while *An. sundaicus*, *An. annularis*, *An. pulcherrimus* and *An. subpictus* are reported as the secondary vectors [28,29]. 

Of 630 districts (an administrative unit in a state having population of around one million or more, Census of India 2011) in India, the major vectors, *An. culicifacies*, *An. fluviatilis* and *An. stephensi* are present in 420, 241, and 243 districts respectively [30]. Such diversity, no doubt, plays a key role in determining the distribution and seasonality of malaria across the country.

*Plasmodium vivax* and *Plasmodium falciparum* are the main malaria parasites, though *P. malariae*, *P. ovale* and *P. knowlesi* have also been recorded, but rarely. *P. vivax* accounts for around half of all malaria cases in the country. As per the available data of 2014, a high Annual Parasite Incidence (API > 5) was found in most of the districts of Chhattisgarh, Jharkhand, Odisha and north-eastern states of India; and a few districts of Madhya Pradesh, Andhra Pradesh, Maharashtra and Gujarat (Figure 1A). Notably, except Andhra Pradesh and some NE states, all of the states, particularly in western part of India, have annual malaria cases less than 50,000, and more cases attributable to *Pv* (Figure 1B), probably because of the phenomenon of relapse/reinfection. The states of Chhattisgarh, Jharkhand and Odisha located in the eastern part are the predominant states for the *Pf* burden, of which the state of Odisha alone shares nearly 50% of this *Pf* burden of the country. The high burden states have the presence of two vectors (*An. culicifacies* and *An. fluviatilis* in the eastern part, while *An. minimus* and *An. dirus* in northeastern states).

The control strategy for malaria is active case detection with blood slides and or rapid diagnostic kits at fortnightly intervals, two rounds of Indoor residual spray (IRS) with appropriate insecticide for vector control and the distribution of Long Lasting Nets in high endemic areas [31].

### 2.2. Climate Model Data

District wise monthly Temperature and RH data for Baseline years (1976–2005) and its near future projection scenario of the 2030s (2021–2040) were extracted from CORDEX South Asia (domain WAS-44i). The near-future projection has an advantage of a minimum climate sensitivity uncertainty that increases in long term projections [32]. CORDEX South Asia downscaled RCM simulations are driven with 10 CMIP5 (Coupled Model Intercomparison Project Phase 5) AOGCMs (Atmosphere-Ocean coupled General Circulation Models) [33], using three of the four greenhouse gas emissions scenarios known as Representative Concentration Pathways (RCPs) [34]. Studies have put forward that the CMIP5 ensemble is able to simulate the broad spatial distribution patterns of the all-India annual mean temperature and precipitation distribution reasonably well [35]. Details on the CORDEX data which are comprised of downscaled climate scenarios for the South Asia region can be available from http://www.cordex.org/.

The CORDEX South Asia dataset includes dynamically downscaled projections to remove systematic error (called bias) and provide a set of high resolution (50 km) regional climate change projections. For the near future projection scenario of the 2030s, Near-Air-surface Temperature and RH, MPI-M-MPI-ESM-LR_MPI-CSC-REMO2009 and MPI-M-MPI-ESM-LR_SMHI-RCA4-simulated climate data models RCP 4.5 scenario, respectively were used. RCP 4.5 corresponds to a low stabilization scenario, and represents the realistic range from a reduction of GHG emissions in the near-term future [36,37]. Increase of global mean surface temperatures under RCP 4.5 is projected to be between 1.1 °C and 2.6 °C (high confidence) until the end of this century with respect to 1986–2005 [38]. The RCP 4.5 was found to be in good coherence with the observed climate of the Indian subcontinent [35] and was, therefore, chosen for the present study. The projected increase in temperature by the 2030s is expected to be 1.7–2 °C. The CORDEX South Asia RCM data sets were downloaded from Earth System Grid Federation (ESGF) Data Node. The other secondary datasets used during model formulation and analysis are observed meteorological data (temperature, rainfall and RH) 2010–2012, and malaria epidemiological data 2013–2017, procured from the Indian Meteorological Department (IMD, Pune) and NVBDCP, respectively.

### 2.3. Selection of Model Indices

The diverse geography of India is characterized by a subtropical monsoon climate. Therefore, to model the climate suitability map for malaria transmission, multicollinearity among temperature, rainfall and RH for 2010, 2011 and 2012 IMD data were assessed. It shows that rainfall and RH are the two predictor variables that are highly correlated (r = 0.8, *p* < 0.05). Thus, both the predictors have a similar effect upon malaria transmission under suitable temperature conditions. Unlike RH, Rainfall pattern (number of rainy days etc.) and quantity is highly variable across the country, and has a lag period effect on malaria transmission [39,40], which also varies geographically. 

The influences of temperature and humidity on the mosquito were considered inseparable in earlier studies [41,42], and were, therefore, considered as indices for the FCSMT model.

### 2.4. Fuzzy-Based Climate Suitability Malaria Transmission (FCSMT) Model Framework

The FCSMT model includes six stages: 1. Generation of monthly interpolated temperature and RH maps for both the periods i.e., Baseline (1976–2005) and projected by 2030s (2021–2040); 2. Determination of fuzzy membership functions for both indices; 3. Creation of fuzzy-monthly temperature and RH suitability maps; 4. Month-wise generation of climate suitability maps for each period; 5. Generation of TW’s-based climate suitability map for both period; and, 6. Generation of ‘climate suitability change map’.

An Inverse Distance Weighting (IDW) [43] interpolation algorithm was applied to further downscale the climate data to a 10 km resolution, the data were later classified into fuzzy subsets by fuzzy membership functions. Fuzzy set theory attempts to generate a consistent representation of an inconsistent reality [44], where the fuzzy membership function (μ) is a curve that defines the degree of belongingness between 0 and 1 [45]. Unlike threshold-based models, fuzzy-based models have an advantage over information or data loss by making it a prime candidate for inclusion in a model during classification.

Vectorial capacity estimates for malaria (relying upon temperature) have generally been represented in Gaussian/binomial shapes [46,47], but in such cases where we work with ‘optimal range’ instead of ‘optimal point’ for Temperature and RH for malaria transmission suitability, these functions are not appropriate. Previous studies suggest that temperature ranging between 18 °C and 32 °C is required for *Pf* transmission [3,48], while for *Pv* it is 16 °C and 32 °C. Regarding RH, the ‘most suitable’ range for transmission is between 55 and 80 percent, while, the lower and upper threshold, which are though poorly delineated so far, are taken as 40 and 95, respectively, on the basis of: 1. Vector survival is least at RH less than 40% [49,50], and 2. Mosquito activity is suppressed with humidity over 95% [51,52,53,54]. Until now, no study has been able to determine the ‘optimal RH%’, as for temp it is 28 °C [48]. Since, we have considered the ‘most suitable range’ for defining temperature and RH suitability thresholds; ‘sinusoidal membership function’ was selected. This function is characterized by four scalar parameters, a, b, c and d. Defined as (Figure 2):

For the temperature index, ‘a’ represents the lower threshold (18 °C for *Pf*, and 16 °C for *Pv*), ‘d’ represents the upper threshold (32 °C for both *Pf* and *Pv*), beyond which the membership function is 0 that is ‘least suitable’; whereas, ‘b’ and ‘c’ represent 24 °C and 28 °C, respectively, this range has been identified as the ‘most suitable’ range for malaria transmission [48,55], the membership function is 1. Similarly, for RH scalar parameters a, b, c and d are 40, 55, 80 and 95, respectively, where the membership function for the range 55% and 80% is 1 as ‘most suitable’, and as the value moves away from the range towards 40% or 95%, the membership function decreases to 0, representing ‘least suitable’ (Table 1).

Arc GIS10 software and its FUZZY tools were used for the FCSMT modeling and mapping, and using a fuzzy intersection operator, monthly climate suitability maps were generated from fuzzy classified monthly temperature maps and RH maps (Refer Appendix A). 

According to Zadeh-intersection, the membership function μ_A∩B_ of the intersection A∩B is point-wise defined for all x ϵ U by: μ_A∩B_(x) = min(μ_A_(x), μ_B_(x). The ‘Composite climate suitability map’, representing the length of transmission windows (TWs), was generated by compiling monthly suitability maps for both baseline and projected 2030s scenarios. A ‘climate suitability change map’ between baseline and projected by 2030s were generated for visualizing the spatio-temporal changes in the length of the TWs, and to identify the new foci of malaria transmission due to climate change.

## 3. Results

### 3.1. Monthly Climate Suitability for Baseline and Projected 2030s for Pv and Pf

Monthly climatic suitability maps (combining temperature and RH) for *Pv* and *Pf* (Figure 3 and Figure 4) show that the northern half of India remains least suitable for malaria transmission in both baseline and projected periods during first six months of the year. The low temperature (below malaria transmission threshold) during the winter months (December, January and February) and high temperature with low humidity (exceeding upper malaria transmission thresholds) during the summer months (March, April and May) are the barriers here. With the onset of the monsoon in the June month, the transmission suitability improves, thereby between July and October about 80% of the country becomes suitable for *Pv* and *Pf* transmission at different suitability scales. The trend remains the same in both baseline and 2030s scenario with some exceptions as projected in the 2030s: 1. Late onset of malaria transmission suitability during monsoon in the Indo-Gangetic plains of India, as the onset of transmission suitability shifts from July to August; 2. Enhancement in malaria transmission suitability during December, January and February months in Southern states of India; 3. Improvement in intensity of transmission suitability in Central and Eastern states between September and November, and 4. For P*v* malaria transmission, a one month window has expanded in the month of November in most of the Central and Southern states. All these projected outcomes indicate the shifting and extension of malaria transmission windows (TWs).

### 3.2. Composite Climate Suitability Map for Baseline and Projected 2030s for Pv and Pf 

The generated composite climate suitability maps for malaria transmission (Figure 5) show the spatial distribution of TWs (no. of months) across the country for both *Pv* and *Pf*. For both the *Pv* and *Pf* malaria, the spatial extent of TWs of 4–6 months and 10–12 months are projected to increase by 2030s. This may result in shifting of some *Pf* areas in the northwestern part of India from 4–6-months category to 1–3 months and some areas in the central part of India from the 7–9 months’ category to the 4-6 months’ category of TWs. Eastern coastal belt, mainly Odisha and West Bengal, are projected to experience increased TWs for *Pv* malaria transmission by the 2030s.

### 3.3. Changes in Climate Suitability between Baseline and Projected 2030s

The climate suitability change map (Figure 6) depicts the changes in number of months of TWs (new foci/extension/reduction) projected to occur by the 2030s as compared to our baseline year. The major changes are: 1. Some areas in a few districts of the western Himalayan states like Jammu and Kashmir, Himachal Pradesh and Uttarakhand, are likely to have new foci of transmission (Table 2); 2. In other Himalayan states like the northern parts of West Bengal, Bihar and Sikkim, a spatial extension of TWs by two months within the districts is more likely; 3. All of the foothill blocks/circles of the Arunachal Pradesh districts are projected to gain up to three months of *Pv* and *Pf* malaria TWs, the highest being the Wakro, Chowkham and Tezu circles of the Lohit district with seven months. 4. Few districts in Rajasthan, Punjab, Gujarat, Maharashtra and West Bengal are also likely to experience nearly two months gain in P*v* malaria TWs, 5. Maharashtra, Madhya Pradesh, Telangana, Odisha, Karnataka and West Bengal are also likely to experience an increase in nearly two TW’s months for *Pf* malaria transmission; 6. Coastal districts of Maharashtra, Odisha, Goa and West Bengal are also likely to become suitable for *Pf* malaria transmission; and, 7. Rest of the districts in the states of Haryana, Bihar, Uttar Pradesh, Andhra Pradesh and Tamil Nadu are projected to experience reduction in number of months of TWs.

## 4. Discussion

Various climate-based models for the suitability of malaria transmission have been proposed around the world in view of climate change for fore-warning and improvement in planning intervention measures [46,56,57,58]. Fuzzy logic has been used in the field of malaria since 1998 [23]. We attempted CORDEX data using fuzzy logic, a soft modeling approach, to map the climate suitability for malaria transmission. Fuzzy sets express how the transition of indices’ suitability from least to most, representing the intensity of suitability, takes place. It offers a better approach to climate suitability classification for malaria transmission than discrete sets. We have generated a Climate Suitability Map for both *Pv* and *Pf* malaria transmission with baseline (1976–2005) and projected 2030s for India using the FCSMT model under the GIS (Geographical Information System) environment, and have projected the changes that are likely to occur in malaria TWs with reference to climate change.

The physiography of the country has a great role in determining the temperature and humidity distribution, which is reflected in the monthly climate suitability maps for malaria transmission. Climate suitability for more than six months ensures stable malaria transmission [3]. In all the southern states, there exists stable climate suitability for malaria transmission, while in northern states monthly suitability decreases considerably from eastern to western parts of the country. Between the two periods (baseline and projected 2030s) for both *Pv* and *Pf* increase in ‘outbreak prone regions’ i.e., for 4–6 months TWs may be registered due to the reduction in ‘stable transmission regions’, and again, 7–9 months of TWs by the 2030s. Meanwhile interior parts of Karnataka, and Telangana are expected to gain 2–3 months of TWs due to a projected increase in temperature in the winter months. Districts in the Himalayan foot hills that are also likely to experience new foci or increase in TWs by 1–3 months, are due to a projected increase in temperature which at present remains below the temperature threshold of malaria transmission. Owing to the variability in temperature as projected during the 2030s, a shift in TWs is also likely in the northern and eastern states. In baseline years, the transmission suitability improves from July onwards and withdraws by October, but in the 2030s scenario, the transmission suitability window extends up to November, thus making these regions vulnerable to prolonged malaria transmission. In short, the projected suitability of malaria transmission by those 2030s in the context of climate change projects the opening of new foci of transmission in the Himalayan region, a change in the length of TWs in most parts of the country including reduction, and a shift in the months of transmission in some regions.

The discussion can further be enriched with an attempt to corroborate the present malaria trend in India with the FCSMT model’s projected outcomes (Figure 7). The western states like Rajasthan (Barmer, Bikaner and Jaisalmer districts) that have less than four months of TWs, have often experienced an outbreak. Malaria seasonality of Saharanpur and Khushinagar, lying at the two extremes of Uttar Pradesh, also corroborates the length of TWs available for transmission, with no change in the number of TWs during the projected 2030s. The situation is similar in Assam and Andaman-Nicobar Islands, where the region is climatically stable and cases are registered throughout the year with peaks during monsoon months. In southern states, stable climate suitability prevails here too, the graphs for Dakshin Kannada (Karnataka), Prakasham (Andhra Pradesh) and Ramanathapuram (Tamil Nadu) show that cases are registered throughout the year, however, their incidence is low. The region has less than 2 API, and even few districts with 0 API. It shows the predominance of the physiographic complexity of the region in terms of soil type, surface gradient, hydro-geology and land use pattern, over climatic suitability. One important fact has to be accentuated that the potential geographic distribution of climate suitability in the study does not directly translate into an actual malaria case scenario. Striking examples are southern states and the state of Punjab. In southern states, the temperature remains mostly between 28 °C and 34 °C, and in spite of climate suitability for 10–12 months, malaria incidence remains low. 

It needs to be evaluated in the light of Mordecai et al. [47], who postulated the optimal temperature for malaria transmission is lower than 32 °C, as suggested earlier [3]. In the state of Punjab, owing to the better economy and health seeking behavior of the inhabitants, the malaria burden is minimal. Studies have suggested that large areas of the United States and Europe are characterized by suitable climates for malaria transmission, yet only few cases occur because of the strength of their public health infrastructure and other economic factors [47].

The near-future malaria transmission patterns, generated from the FCSMT model using one of latest simulated climate database CORDEX and soft partitioning techniques ‘Fuzzy’, have given a broad spectrum of advantages over the projections for malaria transmission that were made in earlier studies [20,40]. First, earlier crisp temperature and RH threshold-based models informed regional climatic suitability in binomial: Yes (1) or No (0); whereas the fuzzy model works on threshold gradation between 0 (least suitable) to 1(most suitable), and thus generates trend-surface maps with third dimensions. The spatial trend-surfaces are indicated by a likelihood of the suitability between 0 (least) and 1 (most), i.e., the closer a region is to value 1, the higher is the suitability intensity, and vise-versa. This is sharply visible for the month of November, where in central India the climate-based transmission suitability is likely to be at its peak during the projected 2030s, while it is weak during baseline. Second, monthly transition of climate suitability (temperature and RH) maps from most to least across the country gives a detailed visualization on the projected changing pattern of malaria transmission. Third, month-wise gradual changes in the climate-based transmission suitability index within the district not only helps in defining the enhancement or reduction in months of TWs at better resolution, but also assists in identifying areas within the district as potential new foci. Fourth, regions with possible reduction in TWs in the projected 2030s have also been identified, which are likely to the possibility of less than six months of transmission and thus suitable for outbreaks of malaria.

Moreover, the climate suitability for district Jaisalmer in Rajasthan shows that except for September, all its months blocked for transmission owing to temperature > 34 °C and RH < 40% in both scenarios, however, the cases occurring from August to October (Figure 6) reflect the effect of the local micro-niche, which makes the transmission possible, but it could not be captured. Otherwise, the CORDEX data-based FCSMT model has efficiently mapped the climatically vulnerable areas of malaria transmission in response to climate change in India. Through the fuzzy-based transmission suitability mapping, month-wise gradual change in the suitability index from least to most within the district has been highlighted, along with the identified few ‘new foci’ for malaria transmission during the 2030s. This soft partitioning approach of malaria transmission suitability mapping has dissolved the crisp boundaries generated from conventional hard classifiers, making the suitability results more close to the reality. Further, in view of the launch of a malaria elimination program in India [31], the gradual increase in malaria outbreak-prone areas due to a reduction in malaria would also necessitate an early warning system for the detection of outbreaks. The projected new foci need health education in the vulnerable communities.

The limitations of the study include: (i) The projected outcome of the FCSMT model are based upon only two climatic parameters (Temperature and RH), and (ii) the certainty of the projected scenario may largely be affected by strengthened intervention measures guided by micro-stratification for malaria control, ecological changes and socio-economic conditions, etc. The outcome of the model can further evolve if intervention measures are included in it. In this study, only the CORDEX model has been used, warranting the need for other climate models as well as epidemiological or vector distribution models for comprehensive conclusions.

The findings of the study would guide the National Program in planning and judicious decision making in new emerging foci and the areas projected to experience increase in transmission months, towards strengthening health infrastructure and making best use of available tools of intervention for controlling of malaria in view of climate change.

## Figures and Tables

**Figure 1 ijerph-16-03474-f001:**
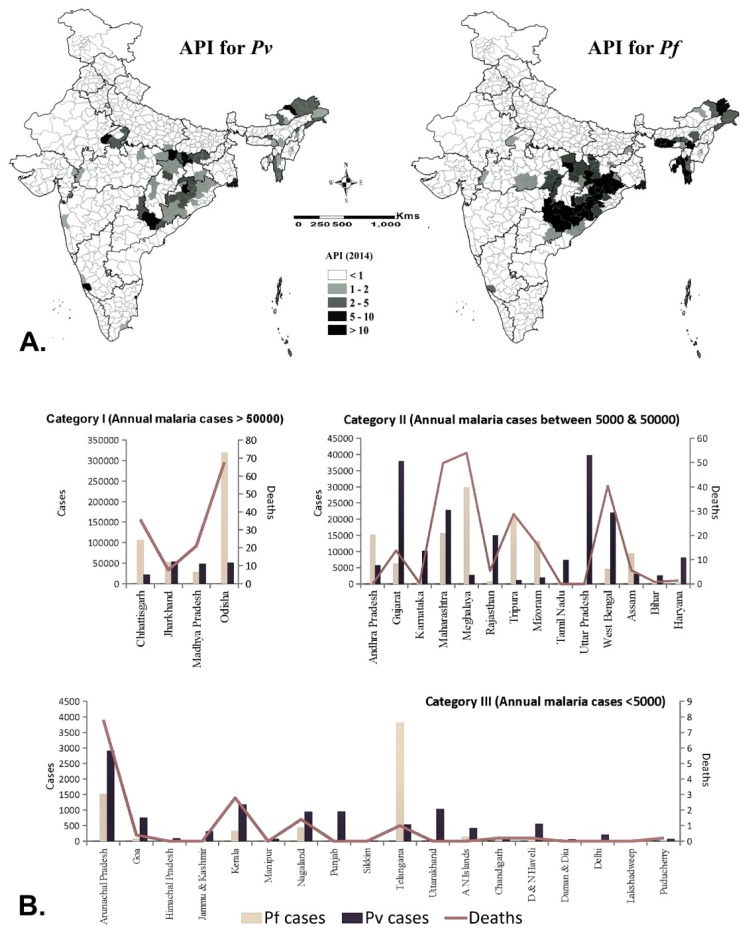
(**A**) Annual parasite incidence (API) for *Plasmodium vivax* (*Pv*) and *Plasmodium falciparum* (*Pf*), 2014; (**B**). State wise average annual distribution of *Pv*, *Pf* and death cases (2013–2017). (Data source: NVBDCP).

**Figure 2 ijerph-16-03474-f002:**
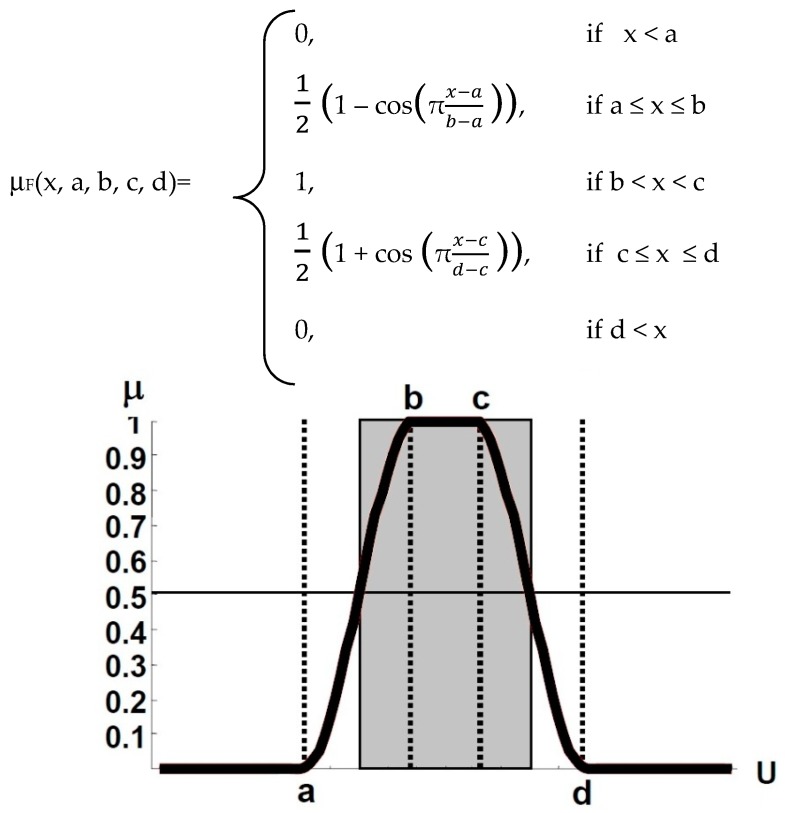
Sinusoidal membership function definition.

**Figure 3 ijerph-16-03474-f003:**
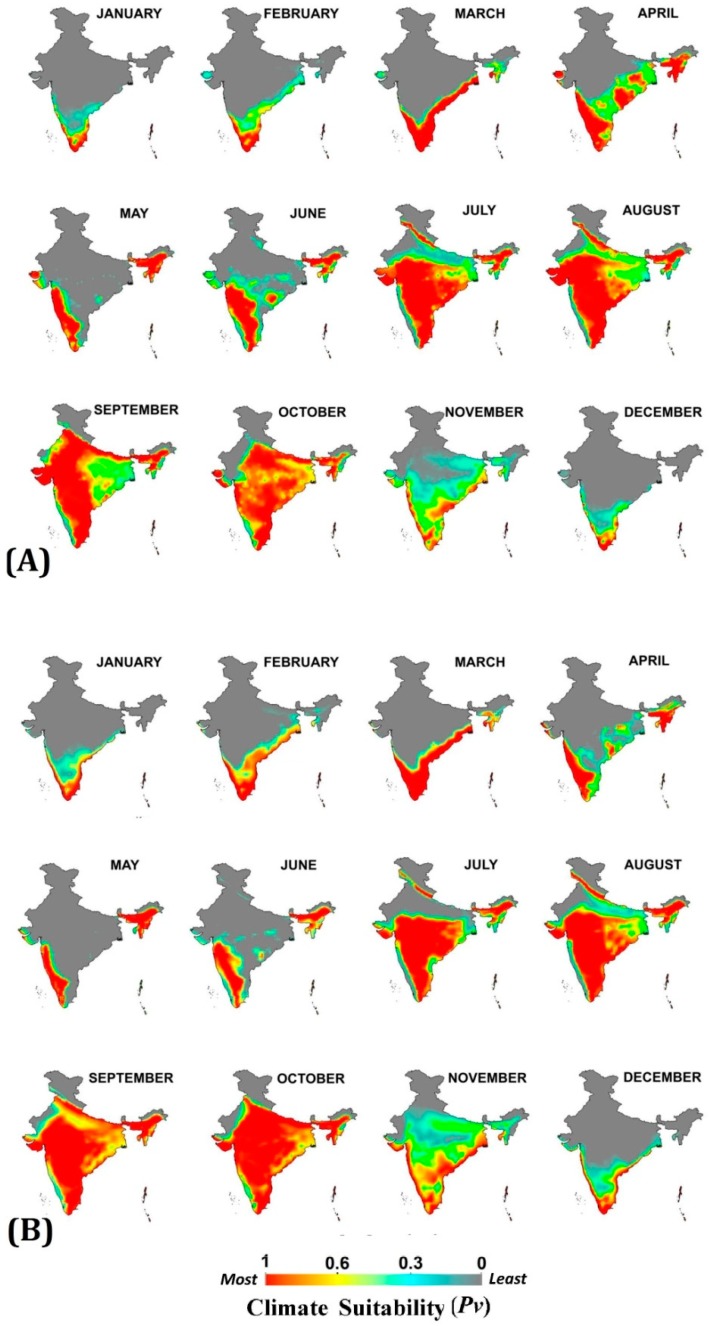
Monthly climate suitability maps for *P. vivax* malaria using Temperature and RH combined (**A**) Baseline (1976–2005) and (**B**) projected 2030s. Increased climate suitability is visible in the months of September to November, while reduction from April to June under the 2030s scenario, as compared to the baseline.

**Figure 4 ijerph-16-03474-f004:**
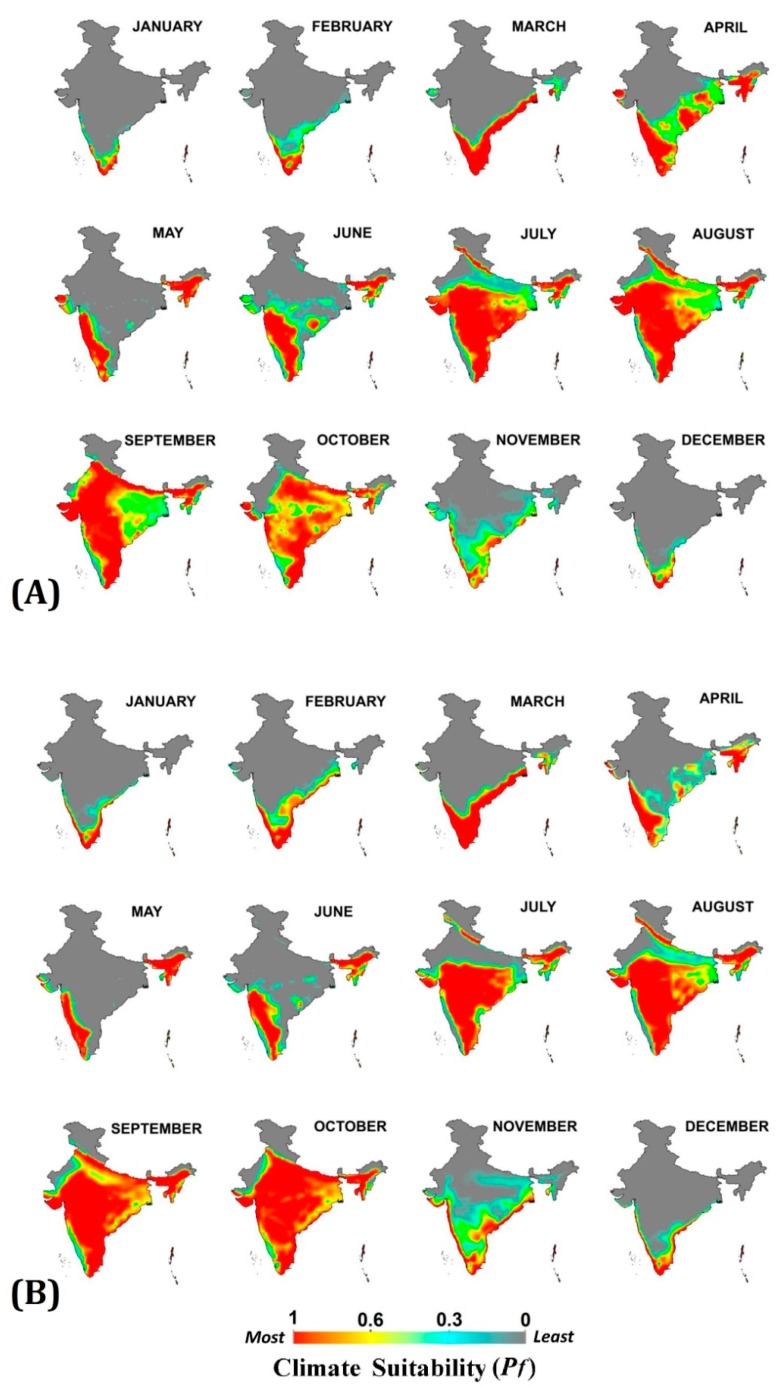
Monthly climate suitability maps for *P. falciparum* malaria using Temperature and RH combined (**A**) Baseline and (**B**) projected 2030s. Increased climate suitability is visible in the months of August to November, while reduction from April to June under the 2030s scenario, as compared to the baseline.

**Figure 5 ijerph-16-03474-f005:**
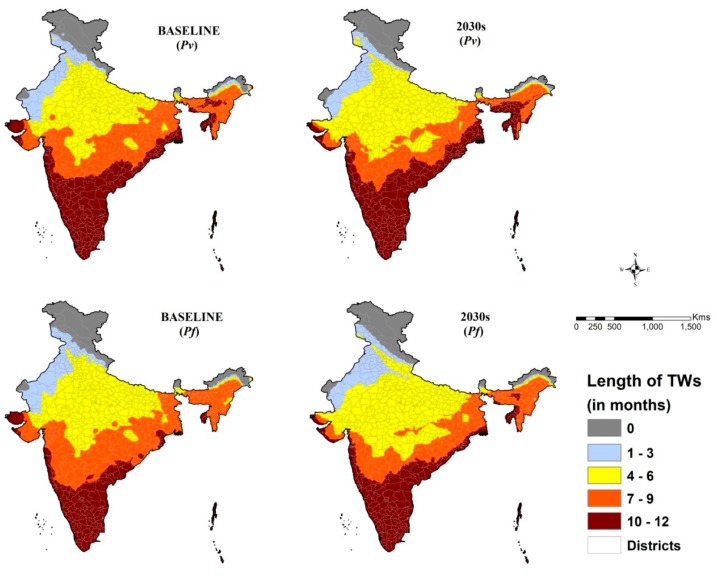
Composite climate suitability map for malaria transmission for the baseline and projected 2030s. For *P. vivax*, new foci of transmission are visible in parts of Jammu & Kashmir, Uttarakhand and Himachal Pradesh. Increase in the number of months of TWs in northeastern states and reduction in parts of Orissa and Gujarat states are also seen. In the case of *P. falciparum*, few foci in Jammu & Kashmir and Uttarakhand; an increase in transmission months in northeastern states and reduction in parts of Orissa and Gujarat states are visible.

**Figure 6 ijerph-16-03474-f006:**
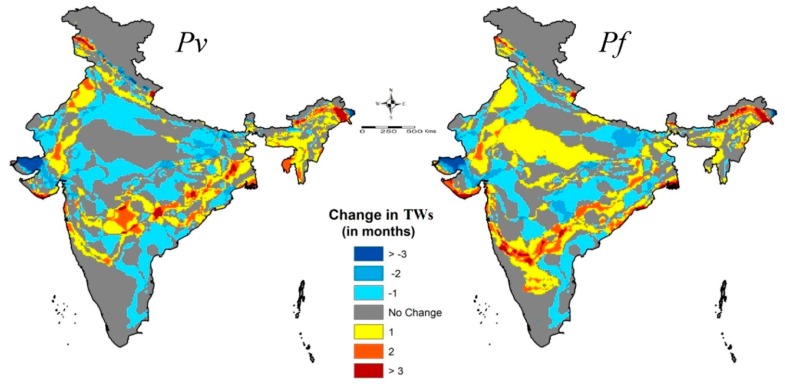
Projected changes in a number of months in the Transmission Window (TW) for malaria (*Pv* and *Pf*) transmission by the 2030s as compared to baseline years. New foci of malaria transmission in the states of Jammu & Kashmir, Uttarakhand and Himachal Pradesh particularly for *P vivax*, while a reduction in parts of Orissa and Gujarat are visible. Increase in the transmission of *P. falciparum* by one month in the central part of India is discernible.

**Figure 7 ijerph-16-03474-f007:**
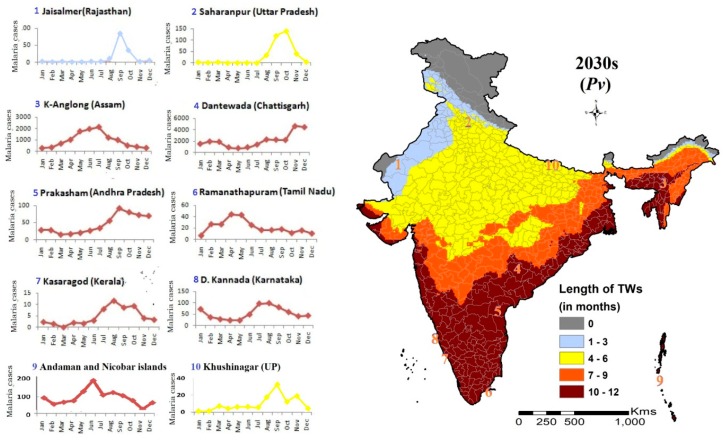
Climate-based Transmission Windows (TW) Vis-à-vis malaria endemicity in selected districts of India. Seasonal fluctuations in 10 representative districts match with months of TWs shown in the map in almost all the districts except Kasaragod (Kerala). In the state of Kerala, malaria transmission is very low due to various other reasons.

**Table 1 ijerph-16-03474-t001:** The scalar parameters of the function.

	Parameters	Temperature (in °C)	RH (in %)	Remarks
Pv	a	16	40	Least suitable
	b	24	55	Most suitable
	c	28	80	Most suitable
	d	32	95	Least suitable
Pf	a	18	40	Least suitable
	b	24	55	Most suitable
	c	28	80	Most suitable
	d	32	95	Least suitable

**Table 2 ijerph-16-03474-t002:** Districts having projected new foci for *P vivax* malaria transmission (TWs in months). (Refer Appendix A).

States	Sl. No.	Districts	Circles/Tehsils	No. of Months	Projected Months
**Jammu and Kashmir**	1	Pulwama	Chrar-e-Shrief	up to 1	July
	2	Srinagar	Shrinagar South	up to 2	July, August
	3	Baramula	Rafaibad, Baramula, Kheeri, Boniar, Tangmarg	up to 2	July, August
	4	Bagdam	All tehsils	up to 2	July, August
	5	Anantnag	Dooru, Kokemag	up to 2	July, August
	6	Shupiyan	All tehsils	up to 2	July, August
	7	Kulgam	All tehsils	up to 2	July, August
	8	Ganderbal	Ganderbal	up to 2	July, August
	9	Ramban	Banihal	up to 2	July, August
**Himachal Pradesh**	10	Chamba	Chamba	up to 2	July, August
**Uttarakhand**	11	Champawat	Lohaghat	up to 2	June, July
	12	Bageshwar	Kapkot	up to 3	June, July, August
	13	Pithoragarh	Berinag, Didihat, Gangolihat, Pithoragarh	up to 4	June, July, August, September

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
