# Peer review of "Shift in Potential Malaria Transmission Areas in India, Using the Fuzzy-Based Climate Suitability Malaria Transmission (FCSMT) Model under Changing Climatic Conditions"

_ijerph, 2019, doi:10.3390/ijerph16183474_

Round 1
Reviewer 1 Report
General assessment and comments:
In the submitted manuscript, Sarkar et al. used a Fuzzy-based Climate Suitability Malaria Transmission (FCSMT) model to predict the impact of climate change on malaria transmission in India. The authors generated the model using the environmental data (temperature and relative humidity) that extracted from CORDEX. From these data, the authors predicted the spatial and temporal changes of malaria transmission in 2030.
This study uses the modeling to address the importance of climate change on malaria transmission in India, which will provide helpful information to the readers in the field who are interested in this topic. Here are some other suggestions for authors to consider.
1. Reference numbers are wrong from 24.
2. In page 5 line 2-3, authors wrote “generate spatially smooth suitability maps for Pv and Pf malaria transmission scenarios for both time periods (Figure 1). ” while Fig.1 only showed data for Pv 2030.
3. All figures need more legends. In the current version, there’s very limited legend (some figures only have title (for instance, Fig 4, Fig5, and Fig. 7)) and it is hard to read the figures without proper figure legends.
Author Response
Point 1. Reference numbers are wrong from 24.
Response1. The correction has been made as pointed out by the reviewer.
Point 2. In page 5 line 2-3, authors wrote “generate spatially smooth suitability maps for Pv and Pf malaria transmission scenarios for both time periods (Figure 1). ” while Fig.1 only showed data for Pv 2030.
Response 2. The section is shifted to supplemental section.
Point 3. All figures need more legends. In the current version, there’s very limited legend (some figures only have title (for instance, Fig 4, Fig5, and Fig. 7)) and it is hard to read the figures without proper figure legends.
Response 3. Needful has been done.
Reviewer 2 Report
Mapping malaria transmission vulnerability in context of climate change in India using Cordex-based climate suitability model
By Soma Sarkar , Vinay Gangare , Poonam Singh , Ramesh Dhiman
Here the authors estimate the change of climatic suitable areas at hazard for malaria transmission in India under current and future conditions using a fuzzy based model. Models to project future developments in vector-borne disease transmission can be of great help to support public awareness and public health planning which is the main idea of this study. However, the submitted manuscript needs to undergo a major revision before supporting policy decision making.
Title
The informative quality should be enhanced. The shift of climatic suitable areas for potential malaria transmission was modelled. Using the term “vulnerability” seems not adequate here. I suggest a title such as: Shift of climatic suitable areas and seasons for potential malaria transmission in India under changing climate conditions.
Abstract:
P1 L15 ff climate data: baseline data is mixed up with RCP4.5 scenario at some points in the manuscript. Describe in a clear way which data was used: baseline (1976-2005) and RCP4.5 for 2030s.
P1 L23 meaning of the sentence unclear: “Furthermore, study has also identified….”. I guess you would like to compare outcomes from figure 2/3 and figure 5: areas with increasing climatically suitability by 2030 show a decrease in transmission months. Please carefully reformulate this sentence.
Introduction:
P2 L12: Literature also talk about an potential decrease in malaria transmission und climate change conditions. https://www.nature.com/articles/srep27771
P2 L19-20: relatively old citations for fuzzy logic based models in the field of vector-borne diseases. There are others published in the meantime please check the literature again. One example:
https://apps.webofknowledge.com/full_record.do?product=WOS&search_mode=GeneralSearch&qid=17&SID=E3a7wnsD9GQRw9L8hNh&page=1&doc=4
Description of malaria vectors in India, distribution of vectors in India, socio economic situation determine transmission in India it missing in the introduction.
Methods:
P2 L33 “2.1 Data” be more specific here.
P3 L3 RCP4.5 is no longer expected to be a realistic scenario in many parts of the world, especially when modelling the near future (2030s), your citations are from 2011 and 2012. Refine your argumentation why you have used (only) RCP4.5. You may add the results for RCP8.5 in the supplemental.
P3 L7 “secondary dataset…various stages of model formulation…” this is not well explained here. I think you have only used this one to estimate the multicollinearity between rainfall and humidity. Please add more detailed information here. Why you did not use cordex baseline to test for multicollinearity between rainfall and humidity?
P3 L29/30 inconsistent line spacing
P3 31 The question arise why you do not use other model types such as epidemiological models or species distribution models (with species = malaria cases), see also https://www.sciencedirect.com/science/article/abs/pii/S1471492217302805
P4 L2ff Add a table to describe the scalar parameters for a better overview: line (a, b, c, d), column (Tpf, Tpv, citation etc.).
P4 L9: Describe what is meant by district level. Perhaps add some information in chapter 2.1.
Is the distribution of vectors somehow limiting the area of potential malaria transmission?
Results:
P4 L19: I suggest to shift text and Figure 1 in a supplemental, because the single parameters are only interesting to a limited degree.
P7 L17: “This may result to reduction in 4-6 months….”. Meaning of the sentence is unclear.
P8 L7ff: When describing the results and with regard to Table 1: either add a geographical map of the districts in the supplemental/method so that readers can refer to, or reformulate your results in terms of: in southern India etc. without referring to districts.
Discussion
The discussion is partially missing to discussing the results, to describe methodological limitations (only some lines P14 L7-10) and to give an outlook (only some lines P14 L10-13). However, shows new data on malaria case/index (Figure 7) which was not introduced in the methods and repeating results (P12 L3ff) in the discussion, or give a justification for the used method in (P13 L8ff) which is expected in the method chapter (and done there: see also p3 L21ff). The discussion need an new overall structure and subheadings.
I think it is difficult to compare API from 2013-2017 with model outcomes of climatic suitability for malaria transmission.
The integration of figure 7 in the result section to validate the model outcomes with regard to TWs seems to be reasonable, although it is limited when the time period (2013-2017?) of the epidemiological data does not overlap with the time period of the model 1976-2005.
P13 L17-19: “… reduction in climatically suitable TWs…”. And “These are climatically vulnerable regions…”. Please check the meaning of the two sentences again. The logical connection is unclear.
Figures:
Add a part © in Figure 2 and 3 where the difference between the baseline and the future projections is shown.
All figure captures should be carefully rewritten, a lot of information is mission here (such as RCP, data sources, years and others)
Figure 2 : Enlarge the single maps by showing 4 months in a lines, and 3 months in a column over the width of the page. Add (B) below (A) and so on for (C). Remove legend in (B) February, and add months (October – December)
Figure 3: Change the format according to figure 2.
Figure 5: Why do you talk about Pv/Pf Scenario. I feel the term “scenario” is misleading in this context.
Figure 7: Add vertical coloured bars in the graphs of the time series when cases occur. The colour should be chosen in accordance with the colours of the maps (e.g. yellow for 4-6 months). Districts for which time series are shown should be better displayed in the map; numbers are only poorly readable.
Overall, I suggest differentiating between spatial and temporal hazard “climate suitable areas for potential malaria transmission” on the one hand and “length of transmission window (in months)” on the other. See figure legend versus figure capture of figure 5.
References:
Numbers are shifted between text and list (by 1).
Carminade NOT Carmindea
Data availability:
Please define here what a “reasonable request” is.
Author Response
Point 1. Title: The informative quality should be enhanced. The shift of climatic suitable areas for potential malaria transmission was modeled. Using the term “vulnerability” seems not adequate here. I suggest a title such as Shift of climatic suitable areas and seasons for potential malaria transmission in India under changing climate conditions.
Response 1. The title has been modified as suggested.
Point 2. Abstract: P1 L15 ff climate data: baseline data is mixed up with RCP4.5 scenario at some points in the manuscript. Describe in a clear way which data was used: baseline (1976-2005) and RCP4.5 for 2030s.
Response 2. correction has been made, as suggested.
Point 3. Abstract: P1 L23 meaning of the sentence unclear: “Furthermore, study has also identified….”. I guess you would like to compare outcomes from figure 2/3 and figure 5: areas with increasing climatically suitability by 2030 show a decrease in transmission months. Please carefully reformulate this sentence.
Response 3. The sentence has been reframed into: ‘Study has also identified the regions with the reduction in transmission months by 2030s leading to unstable malaria having the potential for malaria outbreaks’.
Point 4. Introduction: P2 L12: Literature also talk about an potential decrease in malaria transmission and climate change conditions. https://www.nature.com/articles/srep27771.
Response 4. Literature included in the introduction (Ref No 17).
Point 5.
P2 L19-20: relatively old citations for fuzzy logic-based models in the field of vector-borne diseases. There are others published in the meantime please check the literature again. One example:
https://apps.webofknowledge.com/full_record.do?product=WOS&search_mode=GeneralSearch&qid=17&SID=E3a7wnsD9GQRw9L8hNh&page=1&doc=4.
Response 5. Literature updated.
Point 6. Description of malaria vectors in India, distribution of vectors in India, socio-economic situation determine transmission in India it missing in the introduction.
Response 6. New sub-heading describing the characteristics of malaria in India’ has been included under the methodology section.
Point 7. Methods: P2 L33 “2.1 Data” be more specific here.
Response 7. The section has been modified.
Point 8. P3 L3 RCP4.5 is no longer expected to be a realistic scenario in many parts of the world, especially when modeling the near future (2030s), your citations are from 2011 and 2012. Refine your argumentation why you have used (only) RCP4.5. You may add the results for RCP8.5 in the supplemental.
Response 8. Earlier studies have found good coherence with observed data of Indian subcontinent. Literature added.
Point 9. P3 L7 “secondary dataset…various stages of model formulation…” this is not well explained here. I think you have only used this one to estimate the multicollinearity between rainfall and humidity. Please add more detailed information here. Why you did not use cordex baseline to test for multicollinearity between rainfall and humidity?
Response 9. Secondary data includes both observed meteorological data (temperature, rainfall and RH) 2010-2012 and malaria epidemiological data 2013-2017.
As rainfall was not to be taken for modeling, multicollinearity between rainfall and humidity was not analyzed.
Point 10. P3 L29/30 inconsistent line spacing.
Response 10. Spacing corrected.
Point 11. P3 31 The question arise why you do not use other model types such as epidemiological models or species distribution models (with species = malaria cases), see also https://www.sciencedirect.com/science/article/abs/pii/S1471492217302805.
Response 11. Need for epidemiological models has been mentioned under discussion.
Point 12. P4 L2ff Add a table to describe the scalar parameters for a better overview: line (a, b, c, d), column (Tpf, Tpv, citation etc.).
Response 12. Table added (Table 1).
Point 13. P4 L9: Describe what is meant by district level. Perhaps add some information in chapter 2.1.
Response 13. Mentioned under methodology part 2.1.
Point 14. Is the distribution of vectors somehow limiting the area of potential malaria transmission?
Response 14. Yes.
Point 15. Results: P4 L19: I suggest to shift text and Figure 1 in a supplemental because the single parameters are only interesting to a limited degree.
Response 15. Text and figure shifted to supplementary section (Figure S1).
Point 16. P7 L17: “This may result to reduction in 4-6 months….”. Meaning of the sentence is unclear.
Response 16. P9L8: Sentence is reframed.
Point 17. P8 L7ff: When describing the results and with regard to Table 1: either add a geographical map of the districts in the supplemental/method so that readers can refer to, or reformulate your results in terms of: in southern India etc. without referring to districts.
Response 17. Index map added as supplemental (Figure S2).
Point 18. Discussion
The discussion is partially missing to discussing the results, to describe methodological limitations (only some lines P14 L7-10) and to give an outlook (only some lines P14 L10-13). However, shows new data on malaria case/index (Figure 7) which was not introduced in the methods and repeating results (P12 L3ff) in the discussion, or give a justification for the used method in (P13 L8ff) which is expected in the method chapter (and done there: see also p3 L21ff). The discussion need an new overall structure and subheadings.
Response 18. The identified portion has been reframed.
Point 19. P13 L17-19: “… reduction in climatically suitable TWs…”. And “These are climatically vulnerable regions…”. Please check the meaning of the two sentences again. The logical connection is unclear.
Response 19. Sentence reframed.
Page 15: Fourth, regions with a possible reduction in TWs in projected 2030s have also been identified which are likely to have less than six months of transmission and thus suitable for outbreaks of malaria.
Point 20. Figures: Add a part © in Figure 2 and 3 where the difference between the baseline and the future projections is shown.
Response 20. Legend has been added in Fig 2 and 3 to make it clear about the changes in baseline and projected scenario.
Point 21. All figure captures should be carefully rewritten, a lot of information is mission here (such as RCP, data sources, years and others).
Response 21. Needful has been done.
Point 22. Figure 2: Enlarge the single maps by showing 4 months in a lines, and 3 months in a column over the width of the page. Add (B) below (A) and so on for (C). Remove legend in (B) February, and add months (October – December)??.
Response 22. Corrections have been made as suggested.
Point 23. Figure 3: Change the format according to figure 2.
Response 23. Modified as suggested.
Point 24. Figure 5: Why do you talk about Pv/Pf Scenario. I feel the term “scenario” is misleading in this context.
Response 24. The word ‘scenario’ has been removed.
Point 25. Figure 7: Add vertical coloured bars in the graphs of the time series when cases occur. The colour should be chosen in accordance with the colours of the maps (e.g. yellow for 4-6 months). Districts for which time series are shown should be better displayed in the map; numbers are only poorly readable.
Response 25. Figure modified as suggested.
Point 26. Overall, I suggest differentiating between spatial and temporal hazard “climate suitable areas for potential malaria transmission” on the one hand and “length of transmission window (in months)” on the other. See figure legend versus figure capture of figure 5.
Response 26. Figure title and legend were restructured.
Point 27. References: Numbers are shifted between text and list (by 1).
Response 27. Corrected as suggested.
Point 28. Carminade NOT Carmindea.
Response 28. Corrected as suggested.
Point 29. Data availability: Please define here what a “reasonable request” is.
Response 29. Data sources are already mentioned in the paper and can be obtained/ downloaded freely from the sources.
Word ‘reasonable’ has been removed
Reviewer 3 Report
The control of malaria has done significant progress during the last fifth years.Then, the future challenge of malaria control is to reach the level of disease pre-eliminate and to avoid re-emergence. To reach this objective, additional tools are needed to reinforce the current vector control and chemotherapy's. Modeling the disease evolution could represent a complementary tool for control program to target high risk areas of malaria transmission. Therefore, the study of Sarkar and colleagues may be of interest in regard of this context.
Major remarks
Introduction section: this part of the manuscript must be improved as the authors only describe only the parameters linked to the model developed. It would be helpful to give an insight of malaria transmission and the different foci of Pv and Pf transmission.
In the methodology the section, some climatic and ecological data are missed.
The description of the different eco-geographic areas is important to understand the evolution of several parameters. The authors must show in the methodology the most recent data of malaria transmission across the different areas instead of to show this in the results section.
It would be useful to present the current climate data: range of temperature, RH, rainfall (rainy and dry season), monsoon season. This would help to grasp that there are different climatic or ecological areas, hence a contrast in the transmission of the disease.
Malaria transmission is closely linked to RH, T and specially the rainfall. Why this parameter was not taken into the model? To better predict the evolution of malaria transmission the authors combined RH and T, but which of these to parameter is the best to predict the the evolution by 2030? This is important as the authors excluded or did not show the effect of rainfall.
In the model, the predictions show that the suitability to malaria transmission could increase over time according the period to the year and the region, but what about the prevalence of the disease? How this could evolve over time? Giving this data would strengthen the model. The figure 6 shows the recent API for Pv and Pf but the baseline modeling (fig 2and 3), the suitable areas do not correspond to the areas where malaria transmission is most intense. How to explain this? The model is it enough strength or is lacking some determinant parameters such as rainfall, use of vector control tools...?
In the discussion section the authors must acknowledge the fact that this model could evolve significantly if control strategies are included in it.
And the predictions both at baseline are showing that the most suitable areas are not where malaria transmission is highest. The authors attempted to argue this in 34-38 page 12 but they need to go further as India is endemic for malaria transmission and its control relies essentially on vector control and chemotherapy
Minor remarks
Across the manuscript and based on the literature, optimum RH and T are given for a suitable development of Pv and Pf. It is necessary to give the baseline and 2030 range of T and RH that would be observed in India.
Figure 1: is it referring to the prediction of Pv transmission or for both species?
Figure 3: in the legend it refers to Pf but in the figure it is Pv
In the maps, it is necessary to indicate the several regions that the authors are discussing in the results and discussion sections. It is impossible for the readers who are not familiar with the geographic of India to locate in the maps the different regions the authors are referring.
Author Response
Point 1.
Introduction section: this part of the manuscript must be improved as the authors only describe only the parameters linked to the model developed. It would be helpful to give an insight of malaria transmission and the different foci of Pv and Pf transmission.
Response 1. New sub-heading describing the ‘malaria situation in India’ is included under the methodology section.
Point 2. In the methodology, the section, some climatic and ecological data are missed.
Response 2. All data explained.
Point 3. The description of the different eco-geographic areas is important to understand the evolution of several parameters. The authors must show in the methodology the most recent data of malaria transmission across the different areas instead of to show this in the results section.
Response 3. New sub-heading describing the ‘characteristics of malaria in India’ is included under the methodology section.
Point 4.
It would be useful to present the current climate data: range of temperature, RH, rainfall (rainy and dry season), monsoon season. This would help to grasp that there are different climatic or ecological areas, hence a contrast in the transmission of the disease.
Response 4. New sub-heading describing the ‘characteristics of malaria in India’ is included under the methodology section.
Point 5. Malaria transmission is closely linked to RH, T and specially the rainfall. Why this parameter was not taken into the model? To better predict the evolution of malaria transmission the authors combined RH and T, but which of these to parameter is the best to predict the the evolution by 2030? This is important as the authors excluded or did not show the effect of rainfall.
Response 5. Justification has been given in subheading 2.3.
Point 6.
In the model, the predictions show that the suitability to malaria transmission could increase over time according the period to the year and the region, but what about the prevalence of the disease? How this could evolve over time? Giving this data would strengthen the model. The figure 6 shows the recent API for Pv and Pf but the baseline modeling (fig 2 and 3), the suitable areas do not correspond to the areas where malaria transmission is most intense. How to explain this? The model is it enough strength or is lacking some determinant parameters such as rainfall, use of vector control tools...?
Response 6. The future projections always have limitation. Therefore, the projections may not hold true if strengthened intervention measures are taken which can alter the disease scenario in spite of having climatically most suitability in any area.
The number of months of the transmission shown in Fig 6 match with the malaria endemicity map of India except in southern India. The reason of low endemicity in spite of having suitability for 10-12 months in southern India is a matter of further research to find out whether the optimal temperature for malaria transmission is lower than predicted in the light of model developed by Mordecai et al (2012).
Point 7. In the discussion section the authors must acknowledge the fact that this model could evolve significantly if control strategies are included in it.
And the predictions both at baseline are showing that the most suitable areas are not where malaria transmission is highest. The authors attempted to argue this in 34-38 page 12 but they need to go further as India is endemic for malaria transmission and its control relies essentially on vector control and chemotherapy.
Response 7. Appropriate lines have been added in the discussion as per suggestion.
Point 8. Across the manuscript and based on the literature, optimum RH and T are given for a suitable development of Pv and Pf. It is necessary to give the baseline and 2030 range of T and RH that would be observed in India.
Response 8. Ranges of the potential increase in temperature have been mentioned under methodology section (Increase of global mean surface temperatures under RCP 4.5 is projected to be between 1.1 ÌŠ C and 2.6 ÌŠ C (high confidence) until the end of this century with respect to 1986–2005).
Point 9. Figure 1: is it referring to the prediction of Pv transmission or for both species?
Response 9. Figure 1 highlight the climate suitability for P.vivax only. (shifted to supplemental).
Point 10. Figure 3: in the legend, it refers to Pf but in the figure it is Pv.
Response 10. The correction has been done as suggested.
Point 11. In the maps, it is necessary to indicate the several regions that the authors are discussing in the results and discussion sections. It is impossible for the readers who are not familiar with the geographic of India to locate in the maps the different regions the authors are referring.
Response 11. An Index map for table 2 is given as supplemental.
Round 2
Reviewer 3 Report
The revised manuscript has been significantly improved. This new version provides a better understanding of work and its potential impact in the control of malaria in India